# Impact of Non-Thermal Technologies on the Quality of Nuts: A Review

**DOI:** 10.3390/foods11233891

**Published:** 2022-12-02

**Authors:** Paola Sánchez-Bravo, Luis Noguera-Artiaga, Vicente M. Gómez-López, Ángel A. Carbonell-Barrachina, José A. Gabaldón, Antonio J. Pérez-López

**Affiliations:** 1Laboratory of Fitoquímica y Alimentos Saludables (LabFAS), CEBAS-CSIC, University of Murcia, 25, 30100 Murcia, Spain; 2Department of AgroFood Technology, Miguel Hernandez University, Carretera de Beniel, km 3.2, 03312 Orihuela, Spain; 3Catedra Alimentos Para la Salud, Campus de los Jerónimos, Universidad Católica San Antonio de Murcia (UCAM), 30107 Murcia, Spain; 4Department of Food Technology and Nutrition, Catholic University of San Antonio, Campus de los Jerónimos s/n, 30107 Murcia, Spain

**Keywords:** almond, cold plasma, high pressure, irradiation, mycotoxin, pistachio, pulsed electric field, pulsed light, ultrasound, ultraviolet light

## Abstract

Nuts are widely consumed worldwide, mainly due to their characteristic flavor and texture, ease of consumption, and their functional properties. In addition, consumers increasingly demand natural or slightly processed foods with high quality. Consequently, non-thermal treatments are a viable alternative to thermal treatments used to guarantee safety and long shelf life, which produce undesirable changes that affect the sensory quality of nuts. Non-thermal treatments can achieve results similar to those of the traditional (thermal) ones in terms of food safety, while ensuring minimal loss of bioactive compounds and sensory properties, thus obtaining a product as similar as possible to the fresh one. This article focuses on a review of the main non-thermal treatments currently available for nuts (cold plasma, high pressure, irradiation, pulsed electric field, pulsed light, ultrasound and ultraviolet light) in relation to their effects on the quality and safety of nuts. All the treatments studied have shown promise with regard to the inhibition of the main microorganisms affecting nuts (e.g., *Aspergillus*, *Salmonella*, and *E. coli*). Furthermore, by optimizing the treatment, it is possible to maintain the organoleptic and functional properties of these products.

## 1. Introduction

Nuts are widely consumed worldwide due to their high content of nutrients and bioactive compounds [1]. They can be consumed in different forms and formats, as an appetizer (fresh or toasted), as ingredients (dishes, desserts, ice creams, commercial products, etc.) or as oils [1,2]. Among the great variety of existing dried fruits, the most popular are almonds (*Prunus amigdalis*), hazelnuts (*Corylus avellana*), walnuts (*Juglans regia*) and pistachios (*Pistachia vera*), as well as peanuts (*Arachis hypogea*), which although they are botanically legumes, they are commonly included in the group of dried fruits [2].

Currently, the consumption of nuts is associated with cardioprotective activities (cardiovascular or coronary diseases) and the modulation of other diseases such as cancer [1,3,4]. These activities are mainly due to macronutrients, micronutrients, and other bioactive compounds (proteins, lipid profile, dietary fiber, vitamins, minerals, etc.) that can favorably influence cardiometabolic pathways [2].

However, the consumption of nuts is not risk-free. One of the main risks of consuming nuts is the ingestion of mycotoxin-producing fungi (*Aspergillus flavus*, *Aspergillus parasiticus* and *Aspergillus nonius*, mainly), especially aflatoxins [5]. Aflatoxins cause one of the main risks in agriculture and industry [5,6]. They are carcinogenic, mutagenic, teratogenic, and have immunotoxic properties [7,8,9]. This fungal contamination not only affects human health but also represents an important economic loss in the food industry [8,10,11].

In addition to fungi, there are other microorganisms that can cause alterations in nuts and affect consumers’ health. A clear example is almonds, linked to outbreaks caused by Salmonella in the USA and Canada [12,13]. In addition, similar cases of salmonellosis [14,15,16] and contamination by *Escherichia coli* [17] have been reported in walnuts.

Currently, most of the processes carried out to control these pathogens are based on heat treatments, and most of the consumed foods are subjected to this type of treatment to ensure their safety and extend their shelf life [18]. In nuts, roasting (in oil or air-oven), pasteurization or blanching are the most used heat treatments [19]. However, although these heat treatments are effective in eliminating the microorganisms present in dried fruits, they entails a loss of sensory and nutritional quality [20,21].

On the other hand, consumers today tend towards natural or minimally processed foods, so there is a need to implement alternative methods to heat treatments for food preservation, improving or at least maintaining the nutritional and sensory quality of the processed products [18]. These new technologies have some advantages over heat treatments, such as the use of low temperatures (20–25 °C), which reduces organoleptic losses in foods due to heat treatments [22].

Therefore, this review aims to establish the advantages and limitations of alternatives to thermal processing (e.g., ultraviolet light, plasma treatments, ultrasound, electrical pulses, etc.) used to obtain safe products while preserving their nutritional and sensory quality.

## 2. Non-Thermal Treatments

### 2.1. Ultraviolet Light

UV-C involves the use of light in the spectra range from 200 to 280 nm. Microbial inactivation is caused by the absorption of photons with enough energy to promote a photochemical reaction, resulting in the formation of pyrimidine dimers in the microbial DNA, which impairs cell replication [23].

As mentioned before, UV treatment has increased due to its low cost and easy use in controlling pathogens. However, the inactivation capacity of this technique depends largely on the characteristics of the fruit, such as color or transparency [24], but also the exposure time, microbial species and wavelength [25,26]. In general, the longer the fruit is exposed to UV light and the higher the intensity, the greater the reduction of mycotoxins. In this sense, it has been shown that exposure to ultraviolet-C light in walnuts for 45 min reduced the microbial population of aflatoxin-producing fungi by 4.2 logs, while shorter times (15 and 30 min) reduced contamination by these microorganisms to a lesser extent (1.2 and 3.2 logs, respectively) [27].

In this regard, AFB1 has been shown to be more sensitive to UV radiation at a wavelength of 362 nm due to chemical structural modifications in the terminal furan ring [28]. In peanuts, treatment with ultraviolet radiation (UV-C) at 1080–8370 mJ/cm^2^ showed a reduction of 3.1 log CFU/g of *A. flavous* and a reduction of aflatoxin B1 from 14 to 51% [5]. In addition, intensities of 800 μW/cm^2^ have proven to be effective in peanuts, where 100% of aflatoxin B1 (AFB1) was inactivated with an exposure of 80 min; lower intensities (200 μW/cm^2^) with the same exposure time have presented positive results (approx. 95%) in the inactivation of AFB1 [29]. In pistachios, UV radiation (87.5 μW/cm^2^) decreased the AFB1 content by only 22% after 3 h of exposure and after 13 h, 58% inactivation was achieved [30].

Furthermore, it has been established that 15 min of irradiation with UV at 265 nm was sufficient to completely degrade AFB2 and AFG2 and 45 min was sufficient to obtain a degradation percentage of 96.5 and 100% of AFB1 and AFG1, respectively, in peanuts, walnuts and pistachios after 12 weeks of storage [27]. Babaee et al. [31] also showed that UV-C treatments (12 μw/cm) reduce the aflatoxin amount in pistachios, especially AFB2 and AFG2, without reducing total fat content, protein content, total phenolic compounds or carbohydrates. Likewise, in peanuts, an inactivation close to 100% was achieved after 10 h of exposure at 254 nm [32], and as Babaee, et al. reported [31], without affected the physico-chemical and sensory quality of samples.

Aflatoxin inactivation can also be carried out in nut-based products. For example, in peanut oil, UV radiation under an intensity of 800 mW/cm^2^ for 30 min produced a degradation of AFB1 [33], and UV treatments at 365 nm during 30 min degraded ~96% of AFB1 [26].

UV light treatments have been shown to be effective in inactivating other microorganisms, such as bacteria (*Salmonella* spp., *Escherichia coli* spp., *Listeria* spp., *Yersinia* spp., *Staphylococcus* spp., etc.) [34,35,36]. In shelled walnuts, pulsed UV light at 5, 8 and 13 cm (distances from the quartz window) for 1 to 45 s presented a reduction of *Salmonella enteritidi*, with a maximum reduction of 3.18 log cfu/g at 8 cm during 45 s [37]. Likewise, in peanuts and almonds, a reduction of *Salmonella typhimurium* was observed by applying UV-C irradiation of 10 mW/cm^2^ for 10–30 min [38].

These results are especially important because UV treatments not only offer an inactivation of pathogenic microorganisms but also do not cause significant changes in the physical and sensory quality of the fruits [5,39].

### 2.2. Pulsed Electric Field

Pulsed electric field (PEF) treatment consists of the application of repetitive electric field pulses, generally of high intensity (1–40 kV/cm) for a short duration (μs) and a relatively low energy input to a food placed in direct contact with the electrodes. Its main effect on microorganisms is the permeabilization of the cell membranes [40].

PEF is a non-thermal treatment on which much research has been carried out over the last few years, mainly due to its high capacity for being used to extract beneficial components from industry by-products, such as pressing, drying or osmosis itself. It also reduces the detrimental effects arising from conventional heat treatments. This promising PEF technology is an interesting alternative to other classical methods used, such as cooking or microwaving, it has been tested for the separation, stabilization and dehydration of important compounds without affecting their nutritional properties. In addition to improving the extraction processes and energy costs, with this non-thermal technology, it has been seen that stress can be induced in plant cells, thus stimulating the biosynthesis of bioactive components without causing an environmental impact during their extraction [41,42].

The use of this treatment has been applied in food groups such as fruits, vegetables and nuts due to their high nutritional and antioxidant value, linked to their rich composition of proteins, enzymes, phenols, anthocyanins and carotenoids [43,44,45,46,47].

It has been seen that the use of combined treatments, pulsed electric field (PEF) and ultrasound (US) in the evaluation of bioactive compounds in products such as almonds, gave positive results, leading to the highest contents of phenolic compounds and/or flavonoids [48].

There are no data on nut aromas, although the positive effect of this treatment on the extraction of aroma in plants or the preservation of these aromas in food processing, fruits juices and vegetables has been demonstrated. These changes could be possible due to the fact that PEF induces cell permeabilization, facilitating enzyme-substrate reactions after the treatment [49,50,51,52,53,54].

### 2.3. Pulsed Light

Pulsed light (PL) is a non-thermal technology initially developed to decontaminate foods by killing microorganisms using pulses of a high-power broad-spectrum light, with the UV-C part being the most lethal [55]. PL can heat samples; however, this is a side effect that should be avoided rather than a pursued effect.

The characterization of PL treatments is usually reported in terms of fluence (J/cm^2^), which is the amount of energy imposed on the sample surface divided by the illuminated area of the sample; a more exact definition can be found from Braslavsky [56]. However, this parameter is not always reported, which makes comparison of research findings difficult as has been discussed before [57].

Research about the use of PL in nuts has had three different goals: (i) foodborne pathogen inactivation, (ii) reduction of allergenicity, and (iii) degradation of aflatoxins. Therefore, the potential consequences of PL treatment on the quality of nuts have been studied in the framework of the treatment conditions required to achieve one of these three goals.

In shelled walnuts, the inactivation of *Salmonella* requires 41.2 J/cm^2^ to attain a 3.2 log reduction. Under these conditions, no statistical significant changes (*p* > 0.05) in color parameters (*L**, *a** and *b**) nor in lipid oxidation measured by the malondialdehyde (MDA) method were observed [58]. A deeper study about the effect of subjecting shelled walnuts to a fluence of 41.2 J/cm^2^ on their quality [59] showed that PL indeed caused quality changes in shelled walnut, but these were mild. PL had no statistically significant (*p* > 0.05) effects on lipid oxidation markers, such as concentration of thiobarbituric acid reactive substances (TBARS) and peroxide value, which is consistent with the absence of rancidity perception reported by a trained sensory panel [58]. Nonetheless, PL significantly (*p* < 0.05) increased the concentration of volatile compounds associated with green/herbaceous odors (e.g., 1-hexanol and hexanal) and decreased compounds related to fruity notes (methyl hexanoate) and citrus odors (D-limonene and nonanal) as determined by gas chromatography/mass spectrometry. A descriptive sensory analysis has shown that the descriptors walnut odor and flavor, nut overall taste and aftertaste received statistically significantly (*p* < 0.05) higher scores; the descriptors sweet and woody odor received lower scores, while 16 other traits such as all those related to color, texture, and rancidity were unaffected. Complementarily, the antioxidant activity and total phenol concentration were not significantly affected (*p* > 0.05) by the PL treatment.

In almond kernels, PL has been successfully applied for reduction of the allergenicity of almond protein extracts [60] but requires an estimate of over 150 J/cm^2^, which greatly exceeds the 12 J/cm^2^ maximum allowed by the FDA (1996). This fact, together with the high temperature reached (115 °C), the 58.5% water loss and the circumstance that the efficacy of PL is lower in solid matrixes than in liquid ones led to the conclusion that PL was not appropriate for the treatment of almonds due to its low effectiveness and significant negative impact on almond quality. When used to inactivate *Salmonella enteritidis*, Oner [61] reported a 4.1 log reduction and Harguindeguy and Gómez-Camacho [62] reported a 6 log reduction after application of <50 J/cm^2^, but no effects on almond quality were studied. Finally, Liu et al. [63] used a 1 min immersion in water as a previous step before subjecting almonds to PL to avoid the detrimental effect of excessive heating on dried almonds observed in their preliminary tests. The dipping of almonds in water followed by a PL treatment for 18 min achieved at least a 5 log reduction of *Salmonella*. The treatment slightly affected almond color, as measured instrumentally, but did not significantly increase the lipid oxidation measured by different methods (peroxide value, total acid number, conjugated diene and TBARS), even after storage at 39 °C for 11 d. Furthermore, no significant effect of the treatment on the appearance and color of almonds was visually observed.

In peanuts, PL has been tested to decrease allergenicity and aflatoxin decontamination. The allergenicity of whole kernels was decreased by PL but required fluences >1600 J/cm^2^; the effects of this extreme treatment condition on peanut quality were not reported [64]. PL has also been proven to be able to decrease aflatoxin B1 and B2 concentrations in peanuts with and without skin. No significant (*p* < 0.05) effects were observed when applying >1000 J/cm^2^ in the oil quality indicators (free fatty acid, peroxide value and acidity value) but oil color was affected [65]. A study combining the use of citric acid and PL also decreased aflatoxin B1 and B2 concentrations while affecting peanut color; however, no study of oil quality was undertaken in this case [66].

In general, PL causes mild effects on the quality of nut oils. This is striking since the combination of low water activity, the presence of oxygen and UV light favors the oxidation of unsaturated fatty acids, which are abundant in nuts. The detection of changes in oil quality may require a deeper study beyond the classical oil quality indicators because gas chromatography/mass spectrometry and descriptive sensory analysis have revealed changes are not detected by conventional oil oxidation methods. Color is a quality parameter slightly affected in PL-treated nuts. While PL seemed effective for foodborne pathogenic bacteria inactivation, with only mild quality changes, other applications such as allergenicity abatement and aflatoxin degradation require treatment conditions that negatively affect nut quality.

### 2.4. Ultrasound and High Pressure

Power ultrasound is a part of the sound spectrum ranging from 20 kHz to 1 MHz. Its main inactivation mechanism is acoustic cavitation, which is the formation, growth and implosion of microbubbles within a surrounding medium due to pressure fluctuations induced by ultrasonic waves [40].

For the preservation and quality control of food, this ultrasonic treatment is one of the most investigated, mainly with the objective of reducing the microbial load and inhibiting enzymatic activity, without causing physical–chemical or nutritional changes in food. In addition, this technology usually eliminates microorganisms at levels lower that those reached by thermal treatments and comparable to those reached by current sterilization methods and pasteurization or high pressure [67,68].

It is also used to emulsify oils, sauces and fruit juices and to encapsulate aromas, as this method requires less energy, is economically much more beneficial and improves the efficiency of industrial food processing, including reducing waste and by-products [69,70,71,72].

Regarding the effects of this treatment on the nut aromatic profiles, it has not been investigated to date, although research has been carried out in which sensory improvements in parameters such as aroma and flavor in peanuts were observed [73].

A study in which combined pulsed electric field and ultrasound methods were used suggested an improvement in color and a possible stability of volatile compounds in an almond extract [48].

High pressure processing (HPP) is a technology that uses pressures between 100 and 1000 MPa to treat foods submerged in water and packed in a suitable material. The microbial inactivation by HHP is a complex event related to disturbed cell structure organization and metabolism, including, for instance, cell membrane rupture [74].

Most of the research that has been carried out on the application of high pressures has been directed towards the juice industry and other non-alcoholic beverages [75]. For example, in coconut water, a treatment at 593 MPa for 3 min was effective against *E. coli*, *Salmonella* and *L. monocytogenes* and led to a pleasant sensory profile, similar to that of untreated water [76]. Some recent studies have shown that the application of high pressures (100, 200 and 400 MPa) can even improve the functional properties of pine nut protein [77,78].

On the other hand, the application of HPP has also been analyzed to reduce allergenicity in nuts. Allergic reactions caused by nuts represent 0.5% of the total reactions reported [79]. In this sense, Long et al. [80] showed that applying a treatment of 400, 500 and 600 MPa for 10 min or more in peanuts reduced the immunoreactivity of allergens. Likewise, Hu et al. [81] found that the allergenicity of peanuts decreased with HPP of 60, 90, 120, 150 and 180 MPa. HPP has also been effective in reducing allergens present in soybeans [82,83]; treatments of 300 MPa for 15 min reduced allergens in soybeans and soybean sprouts [83] and in isolated soy protein for baby foods [82].

### 2.5. Irradiation

γ-rays are high-energy photons produced from radioactive isotopes such as cobalt-60 and cesium-137, with an average total dose equal to or less than 10 kGy. They affect microorganisms and food matrixes by direct disruption of biomolecules such as DNA and by indirect effects due to the interaction between biomolecules and radicals and ions originating from water radiolysis [84]. Irradiation treatments could penetrate through the shell of the nuts and give a homogenous treatment [85].

Directive 1999/3/EC [86] of the European Parliament and of the Council, on the establishment of a community list of foods and food ingredients authorized for treatment with ionizing radiation, establishes a single category of ingredients that can be irradiated throughout the European Union: aromatic herbs, spices and vegetable seasonings (dry). However, Member States were allowed to continue irradiation in other categories authorized before the entry into force of the directive. For this reason, there are currently EU countries where the irradiation of other types of food, such as nuts, still takes place.

The commercial application of irradiation is growing globally, primarily in Asia and in the Americas. However, the trend in the European Union is the opposite, with fewer products being irradiated. In general, irradiation technology is not well understood by the consumer, who tends to have a negative perception of this treatment [87].

One of the problems with nuts is that they can be contaminated by pest damage, mold and yeast, with some of them forming the dangerous aflatoxins (*Aspergillus flavus* and *Aspergillus parasiticus*), which decrease both the quality and shelf life of the nut and are carcinogenic [88,89,90]. Doses of 20 kGy have proven to be effective in reducing these molds on walnuts (Food Irradiation research technology). In peanuts, doses of 5 kGy and higher significantly decreased (89% approximately) the growth of *A. parasiticus*; however, they did not achieve its complete elimination [91]. In addition, in peanuts, 6 kGy detoxified around 75% of aflatoxin B1 [92].

Microbial pathogens (*Salmonella* and *E. coli*) in nuts are also one of the recent problems associated with these products [12,13,14,15,16,17]. Irradiation has been shown to be effective against most food pathogens (*Campylobacter jejuni*, *Aeromonas hydrophia*, *Yersinia enterocolitica*, *Salmonella*, *Shigella*, *E. coli*, *Listeria monocytogenes*, and *Staphylococcus aureus*), which have a low tolerance to it, especially in nuts, due to low moisture [85]. Doses between 3 and 5 kGy produce a 4 log cfu/g reduction of *Salmonella* in almonds [93] and reduce, below the detection limit (1 log cfu/g), the concentration of *E. coli*, *Salmonella Thyphimurium* and *L. monocytogenes* in pistachios [94].

Nuts are very susceptible to lipid peroxidation, and therefore, off-flavors and rancidity could be formed during the processing and shelf life [95,96].

The first sensory tests, to determine the formation of off-flavors, in dry fruits such as raw almonds or pistachios, were carried out by O’Mahony et al. [97] and Thomas [98]. No differences were observed in the rancidity parameter, up to irradiated doses of 10 kGy and up to 6 months of storage. Subsequently, Wilson-Kakashita et al. [99] treated walnuts with gamma irradiation doses of up to 20 kGy and did not find any sensory differences in the previously mentioned parameters of off-flavors and rancidity.

Mexis and Kontominas [100] irradiated cashew nuts up to doses of 7 kGy and found that the concentration of volatile compounds, such as alcohols, ketones and aldehydes, increased after being treated, although sensory analysis showed that cashew nuts remained organoleptically acceptable at doses below 3 kGy. Taipina et al. [101] also obtained the same results at irradiation doses of up to 3 kGy in pecan nuts regarding sensory parameters such as aroma and flavor attributes.

With higher irradiation doses up to 7 kGy in pine nut kernels, almonds and hazelnuts, with storage periods of up to 6 months, no sensory differences in flavor were detected [102,103,104]. Sánchez-Bel et al. [104] reported that irradiation doses up to 7 kGy caused no significant changes in sensory quality.

Recent studies on pistachios have analyzed the evolution of volatile compounds and related sensory parameters, such as taste, at doses of up to 2 kGy; 14 volatile aromatic compounds were found in non-irradiated pistachios. The main compounds at doses lower than 1.5 kGy were terpenes; however, at doses ≥ 2 kGy, they were aldehydes, terpenes and alcohols. The amounts of volatile compounds increased in the irradiated product, up to 21; 4 aldehydes and 3 more alcoholic compounds were identified, and these increased with the irradiation doses (≥4 kGy), showing a significant difference (*p* < 0.05). Aldehydes comprised approximately half of the volatile compounds in the sample irradiated at 6 kGy. Aldehyde components, including heptanal, octanal, nonanal and 2-nonenal, and alcohols comprising 1-hexanol, octanol and 1-nonanol were formed during irradiation, as they did not exist previously. Values ranged from 5.95 mg/kg for non-irradiated nuts to 14.97 mg/kg of total volatile compounds for pistachios irradiated at a dose of 6 kGy, for heptanal, with a green sweet taste, and hexanal, an indicator of oxidation and quality of fatty foods, which is one of the main components of irradiated nuts that can be derived from linoleic acid (Frankel, 1983). α-pinene was the most abundant volatile compound in all the samples, as shown by other studies on this dried fruit [105]. The taste of samples was affected (*p* < 0.05) at doses ≥ 2 kGy, obtaining the worst results as the irradiation dose increased to 6 kGy [106].

The authors reported that the oxidation of unsaturated lipids (e.g., unsaturated fatty acids) as a result of irradiation could produce a variety of volatile compounds, including aldehydes, ketones, alcohols and esters, which can significantly affect the organoleptic properties of foods in extremely small quantities [100,107].

### 2.6. Cold Plasma

Plasma is universally known as the fourth state of matter. It is produced by the ionization of gas particles by adding energy, with the consequent generation of ions, free radicals, molecules in an excited state, and photons [108]. Cold plasma is achieved by generating high electrical discharges at room temperature (or similar), achieving ionic destabilization of the particles [109]. Reactive species such as O, O_3_, NO, NO_2_, and OH generated by plasma lead to microorganism inactivation and cellular deformation [110]. Applied to food, these electrical discharges destabilize cells, causing membrane damage, protein denaturation, lipid peroxidation, and DNA and/or RNA damage, among others [111]. In addition, it is a technique that does not leave any type of residue and does not raise the temperature of the treated product; thus, it does not greatly affect its original properties [112]. For this reason, cold plasma is being widely investigated for its use as an antimicrobial agent in food, antigerminative, toxin inactivator and biofilm control.

In the case of nuts, the influence of cold plasma as an inactivator of microorganisms has been widely studied in almonds (Table 1). Deng et al. [113] achieved reductions of 5 log cfu/g of *E. coli*, using 30 s treatments with a voltage of 30 kV and 2 kHz. Hertwig et al. [114] also studied the effect on almonds in the case of *Salmonella enteritidis* PT30 and using whole almonds. They studied the application of cold plasma with air at room temperature (20 kV and 15 kHz) and achieved reductions of more than 5 log cfu/g. In addition, in *Salmonella,* Khalili et al. [115] studied the effect of plasma (in addition to *Shigella*), obtaining for both reductions of more than 4 log cfu/g using the gliding arc plasma technique (14 kV, 50 kHz and 4 min of treatment). Shirani et al. [116] studied the effect of plasma on peeled and sliced almonds by applying 17 V and 2.26 A current for different time periods (5, 10, 15, and 20 min). By using 20 min argon cold plasma jet treatment, the total count of almond microorganisms was reduced 2.95 log cfu/g, mold and yeast 1.81 log cfu/g, and *Staphylococcus* aureus 2.72 log cfu/g. Furthermore, almond color, peroxide value and sensory evaluation did not change with this treatment.

In pistachio (Table 1), cold plasma has been studied, mainly for its use in the control of *Aspergillus*. Several authors have shown that this technique is very effective for this microorganism. Makari et al. [117] found that after 3 min of exposure to a power of 130 W, 20 kHz frequency and 15 kV voltage, the spores were undetectable (reduction of 4 log cfu/g). They also reported a reduction in the AFB1 content by up to 60%, without affecting fruit quality. These authors show that the longer the exposure time, the greater the antimicrobial effect achieved. Ghorashi et al. [118] treated whole pistachios using three types of plasma: capacitive coupled plasma (AP-CCP), direct current diode plasma (DC-DP), and inductively coupled plasma (ICP), with the DC-DP being the most effective in the *Aspergillus flavus* reduction rate (5-log reduction; 83%) at 300 W input power, 2 Torr pressure and 20 min irradiation. Tasouji et al. [119] studied the effect of cold plasma, in this case using an Atmospheric Pressure Capacitive Coupled Plasma (AP-CCP) generating device using argon gas, in the reduction of *Aspergillus flavus* in pistachio. These authors obtained a reduction of 4 log fungus reduction (about 67%) by using a power input of 100 W for a 10 min irradiation time, with no alteration on the nut texture. The effect of this technique in pistachios on the decontamination of *Aspergillus brasiliensis* and *E. coli* has also been studied. Pignata et al. [120] used low-pressure gas plasma with three gases: pure argon, pure oxygen and argon-oxygen (50%), with the latter being the most effective. A treatment for 5 min against *Aspergillus brasiliensis* achieved a reduction of 5 logs, and 4 logs in *E. coli,* by applying the treatment for 1 min for decontamination of pistachios [120]. The study demonstrated that a mixture of argon-oxygen-generated plasma was more effective for decontamination than pure oxygen and pure argon. The treatment time of 30 min led to a reduction of 3.5 logs of *Aspergillus brasiliensis* using pure oxygen and pure argon. However, a treatment time of 5 min, 1 min and 15 s led to a reduction of 5.4 logs using an argon and oxygen gas mixture. In *E. coli*, 1 min and 30 s of treatment resulted in 4 log reductions for oxygen and argon, respectively.

This technique has been studied in other nuts, such as peanuts (Table 1). Lin et al. [121] achieved reductions of more than 5 log cfu/g of *Aspergillus flavus* and *Aspergillus niger* using cold plasma with argon (200 W and 5 min of treatment). In addition, the authors specify that microscopically it is observed how this treatment seriously damaged the aflatoxin spores. In walnuts (Table 1), Ahangari et al. [122] observed reductions in the entire content of coliforms and mold through the application of cold plasma (50 W and 20 min). However, in this case, the treatment caused a color change in the samples (darkening).

Cold plasma is an area in which considerable research is being conducted. From what has been shown so far, in the case of nuts, it is a technique that works well as an antimicrobial. It is an inexpensive technique, which has a low environmental impact and which preserves the physical and functional properties of dried fruits. Therefore, it is an emerging technology that deserves to be considered as an alternative in the preservation of nuts. However, research should be focused on obtaining the best combination of processing parameters to achieve the best results.

**Table 1 foods-11-03891-t001:** Effect of cold plasma in nuts.

Nut	Processing Conditions Applied	Effects	Reference
Almond	Non-thermal plasma (NTP); 30 s; 30 kV; 2000 H	5-log reduction of *E. coli.*	[113]
	Cold plasma (CP); air; 20 kV; 15 kHz; 15 min	5-log reduction of *Salmonella enteritidis* PT-30	[114]
	NTP; Gliding arc plasma; 14 kV; 50 kHz; 4 min; 6 mm distance	4-log reduction of *Salmonella* and *Shigella*	[115]
	CP; 17 V, 2.26 A; 20 min; 2 cm distance; ambient temperature	3-log reduction of total microorganisms; 1.8-log reduction of molds and yeasts; 2.7-log reduction *Staphylococcus aureu*	[116]
Pistachio	CP; 130 W; 20 kHz; 15 kV; 3 min; ambient temperature	4-log reduction *Aspergillus*; AFB_1_ reduction of 60%	[117]
	Direct current diode plasma (DC-DP); 300 W; 2 Torr; 20 min	5-log reduction *Aspergillus flavus*	[118]
	Atmospheric pressure capacitive coupled plasma (AP-CCP); argon; 100 W; 10 min; ambient temperature	4-log reduction *Aspergillus flavus*	[119]
	Low pressure cold plasma (LP-CP); O_2_-Ar, 5 mbar; 20 mm distance; 50 MHz; 400 W; 5 min; ambient temperature	5-log reduction *Aspergillus brasiliensis*; 4-log reduction *E. coli*	[120]
Peanut	CP; Ar; 200 W; 5 min; 1 atm; 4 cm; ambient temperature	5-log reduction *Aspergillus niger* and *Aspergillus flavus*	[121]
Walnut	CP; 500 mTorr; 13.6 MHz; 50 W; 20 min	Total reduction in molds and coliforms	[122]

## 3. Conclusions and Future Trends

Non-thermal technologies are proving to be efficient in the antimicrobial treatment of nuts. These technologies (cold plasma, high pressure, irradiation, pulsed electric field, pulsed light, ultrasound, and ultraviolet light) have a low environmental impact and a low economic cost; thus, their application can revert directly to the industry and the consumer. Ultraviolet light is effective against mycotoxins, especially those of the aflatoxin group. Pulsed electric field and the application of ultrasound provide good results due to the fact that they cause an increase in the functional properties of nuts. Pulsed light, irradiation and cold plasma have been effective in reducing *Salmonella*, *E. coli* and *Aspergillus*, without affecting the quality of the treated fruit. High-pressure methods are currently in the development phase in nuts because of their great effect on liquid foods. Currently, this technique is being used to study its effect against the inactivation of allergens in food. Non-thermal treatments are proving to be effective for the microbial reduction of nuts; however, to date, tests have been developed with a small amount of product and have not yet been extrapolated to an industrial level for the treatment of large amounts. In the near future, research should focus on the effect against the allergic power that nuts have in a significant part of the population, because some of the non-thermal treatments have achieved promising advances.

## Data Availability

Data is contained within the article.

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
