# Peer review of "Impact of Non-Thermal Technologies on the Quality of Nuts: A Review"

_foods, 2022, doi:10.3390/foods11233891_

Round 1

Reviewer 1 Report

Dear Authors,

Due to their properties, nuts are very popular all over the world. Consumers expect as little processed as possible. The use of non-thermal preservation methods is a very promising field for nuts, especially in a microbiological context. They also show promise in reducing energy costs.

The prepared mauscript is well-organized and clearly describes the use of the latest non-thermal techniques in preserving the nuts. The tables are very clear. 85% of the literature contains items from the last 10 years.

Only the literature list should be revised in line with Foods requirements.

minor revision.

Author Response

Dear Editor and reviewers

We would like to express our sincere appreciation for the time and effort spent by the two reviewers in the evaluation of our manuscript titled “Impact of non-thermal technologies on the quality of nuts. A review.” [Foods] Manuscript ID: foods-2057079 - Minor Revisions. Thank you very much for your positive and constructive comments and suggestions, which are very helpful for the improvement of our paper. We have considered the comments carefully and have revised the manuscript thoroughly based on the comments. We deeply appreciate your work, and hope that the corrections and responses will meet with your approval. Revised portions are marked, in different colour, in the revised manuscript and the point-to point responses to the comments are listed below in this cover letter. We look forward to your information about our revised paper.

Best regards,

Yours sincerely,

Prof. Dr. Antonio José Pérez-López

REVIEWERS COMMENT

  1. Dear Authors,

Due to their properties, nuts are very popular all over the world. Consumers expect as little processed as possible. The use of non-thermal preservation methods is a very promising field for nuts, especially in a microbiological context. They also show promise in reducing energy costs.

The prepared mauscript is well-organized and clearly describes the use of the latest non-thermal techniques in preserving the nuts. The tables are very clear. 85% of the literature contains items from the last 10 years.

Only the literature list should be revised in line with Foods requirements.

minor revision.

Done as suggested. Literature list has been revised according to Foods journal. Endnote 2018 software has been used.

Reviewer 2 Report

Dear authors,

I have read the article foods-2057079 - Impact of non-thermal technologies on the quality of nuts. A review, and I think it needs small improvements.

I suggest you complete subchapter 2.5 with information on:

- the mode of action of irradiation on microorganisms in nuts,

- the species that are affected by this treatment,

- the level of decrease in mycotoxin concentrations,

- the degree of reduction of microorganisms.

I select these clarifications because the conclusions show the fact that irradiation reduces the level of Salmonella, but without finding any clarification in subchapter 2.5.

Thus, subchapter 2.5 will have to be completed and reconciled with the conclusions.

Author Response

Dear Editor and reviewer

We would like to express our sincere appreciation for the time and effort spent by the two reviewers in the evaluation of our manuscript titled “Impact of non-thermal technologies on the quality of nuts. A review.” [Foods] Manuscript ID: foods-2057079 - Minor Revisions. Thank you very much for your positive and constructive comments and suggestions, which are very helpful for the improvement of our paper. We have considered the comments carefully and have revised the manuscript thoroughly based on the comments. We deeply appreciate your work, and hope that the corrections and responses will meet with your approval. Revised portions are marked, in different colour, in the revised manuscript and the point-to point responses to the comments are listed below in this cover letter. We look forward to your information about our revised paper.

Best regards,

Yours sincerely,

Prof. Dr. Antonio José Pérez-López

REVIEWER COMMENTS

Reviewer #2:

Dear authors,

I have read the article foods-2057079 - Impact of non-thermal technologies on the quality of nuts. A review, and I think it needs small improvements.

I suggest you complete subchapter 2.5 with information on:

1.- the mode of action of irradiation on microorganisms in nuts. Done as suggested, please, see lines 262-263

2.- the species that are affected by this treatment. Include as suggested. Please, see lines 276-277 and 285-286.

3.- the level of decrease in mycotoxin concentrations. Include as suggested. Please, see lines 276-282.

4.- the degree of reduction of microorganisms. Include as suggested. Please, see lines 276-277 and 283-290.

I select these clarifications because the conclusions show the fact that irradiation reduces the level of Salmonella, but without finding any clarification in subchapter 2.5.

Thus, subchapter 2.5 will have to be completed and reconciled with the conclusions.

Done as suggested.